# Treatment of Periodontal Infections, the Possible Role of Hydrogels as Antibiotic Drug-Delivery Systems

**DOI:** 10.3390/antibiotics12061073

**Published:** 2023-06-19

**Authors:** Adelaide Mensah, Aoife M. Rodgers, Eneko Larrañeta, Lyndsey McMullan, Murtaza Tambuwala, John F. Callan, Aaron J. Courtenay

**Affiliations:** 1School of Pharmacy and Pharmaceutical Sciences, Ulster University, Coleraine BT52 1SA, UK; mensah-a@ulster.ac.uk (A.M.); j.callan@ulster.ac.uk (J.F.C.); 2The Wellcome-Wolfson Institute for Experimental Medicine, Queen’s University Belfast, 96 Lisburn Road, Belfast BT9 7BL, UK; aoife.rodgers@qub.ac.uk; 3School of Pharmacy, Queen’s University Belfast, 96 Lisburn Road, Belfast BT9 7BL, UK; e.larraneta@qub.ac.uk; 4DJ Maguire and Associates, Floor 1, Molesworth Place, Molesworth Street, Cookstown BT80 8NX, UK; lyndseymcmullan@hotmail.com; 5Lincoln Medical School, Universities of Nottingham and Lincoln, Brayford Pool Campus, Lincoln LN6 7TS, UK; mtambuwala@lincoln.ac.uk

**Keywords:** periodontal disease, antimicrobial resistance, antibiotic therapy, hydrogel, dental, drug delivery

## Abstract

With the advancement of biomedical research into antimicrobial treatments for various diseases, the source and delivery of antibiotics have attracted attention. In periodontal diseases, antibiotics are integral in positive treatment outcomes; however, the use of antibiotics is with caution as the potential for the emergence of resistant strains is of concern. Over the years, conventional routes of drug administration have been proven to be effective for the treatment of PD, yet the problem of antibiotic resistance to conventional therapies continues to remain a setback in future treatments. Hydrogels fabricated from natural and synthetic polymers have been extensively applied in biomedical sciences for the delivery of potent biological compounds. These polymeric materials either have intrinsic antibacterial properties or serve as good carriers for the delivery of antibacterial agents. The biocompatibility, low toxicity and biodegradability of some hydrogels have favoured their consideration as prospective carriers for antibacterial drug delivery in PD. This article reviews PD and its antibiotic treatment options, the role of bacteria in PD and the potential of hydrogels as antibacterial agents and for antibiotic drug delivery in PD. Finally, potential challenges and future directions of hydrogels for use in PD treatment and diagnosis are also highlighted.

## 1. Introduction

The term periodontal disease (PD) is an umbrella term used to describe chronic inflammatory conditions of the soft tissues or supporting tissues surrounding the teeth [1,2]. In its severe form, PD, a biofilm-induced chronic inflammatory condition of the tissues supporting the teeth, affects more than 700 million individuals worldwide [3,4].

PD is known to begin with the inflammation of the gingiva. This inflammatory response is often initiated by bacteria in a biofilm surrounding the teeth and dental supporting tissues in the form of plaque [5,6]. In circumstances of uncontrolled inflammation, there is associated damage to teeth and supporting tissues, as illustrated in Figure 1.

Periodontal infections are a significant cause of tooth loss and can have negative impacts on overall health. It is not surprising that several studies have identified a bidirectional association between PD and other systemic conditions such as coronary heart disease [8], cognitive disorders [9], diabetes [10], and other cardiovascular diseases [11]. Owing to the debilitating effect of PD, it is known to have a significant impact on the health status and quality of life of individuals.

To prevent irreversible damage to the gingiva, bone, ligament and subsequent tooth loss, several approaches that are aimed at either removing or offsetting the microbiome that makes up the offending plague or ameliorating the inflammatory response have been investigated. The prevention and treatment of PD involve both pharmacological and non-pharmacological options to achieve optimum care.

In the pharmacological management of PD, various treatment options are available for medical emergencies in dental practices including odontogenic pain, viral infections, fungal infections and bacterial infection [12]

In PD, it is crucial to receive treatment following bacterial infections; failing to do so might result in complicated oral and systemic health problems. In the treatment of periodontal bacterial infections, different antibiotics available as various formulations and dosage forms are used by dental practitioners. Most bacterial infections can be effectively managed by the rational use of antibiotics. However, conventional antibiotics are losing their effectiveness against certain bacterial infections due to increases in antimicrobial resistance (AMR).

Challenged by the threat of AMR, researchers have been diligently exploring the potential of advanced antibacterial compounds or materials to address the rising dangers posed by drug-resistant bacteria. Among these, natural extracts, nitric oxide, transition metal dichalcogenides (molybdenum disulfide as an example [13]) and metals ions such as copper [14], silver nanoparticles [15], polymers and polymeric systems have been explored individually in drug-delivery systems or as combination therapy [16]. It is, therefore, imperative to use innovative antibacterial materials and drug-delivery systems to address this growing concern of AMR in PD management.

Several antibacterial polymers were reported approximately 20 years ago [17,18]. Presently, antibacterial polymers and polymeric systems have proven to be effective against clinically relevant bacteria [19,20]. Hydrogels in a polymeric system can be fabricated from different polymers with inherent antibacterial activities, which in effect, confers specific antibacterial activity of the hydrogel formulation. Depending on the polymers used, hydrogels could contain hydrophobic, cationic and amphiphilic moieties that make the formulations attractive alternatives to conventional antibiotics with a lower tendency of AMR [21].

These hydrogels have been proven to have mechanisms of action that are different to most conventional antibiotics and hence have great potential as antibiotic agents [22,23]. Hydrogels as antibiotic drug-delivery materials or antibiotic agents could play a significant role in the global efforts to fight AMR. In recent years, hydrogel-based antibiotic drug-delivery devices have shown potential in the treatment of periodontal infections.

Several works have reported on the antibacterial activity of hydrogels in early-stage in vitro experiments using minimum inhibitory concentrations (MIC) and zones of inhibitions (ZOI) with a number of reports on antibacterial effects in vivo [24,25,26,27]. To date, the translation of antimicrobial hydrogels into therapies for PD has been hampered by limited exploration. Another hindrance is limited data on drug release profiles of such formulations and scarce information on antibacterial selectivity.

## 2. Role of Bacteria in the Prognosis of Periodontal Disease

The aetiology of PD is dental plaque that consists of microbial biofilm that adheres to the tooth’s and gingiva’s surfaces [28]. This biofilm is composed of various bacterial species, and their interaction with the host immune response ultimately determines the prognosis and severity of the disease. In recent years, research has focused on identifying specific bacterial species that are associated with PD and understanding their role in disease progression.

One of the most well-known groups of bacteria associated with PD is the red complex, which includes *Porphyromonas gingivalis*, *Tannerella forsythia*, and *Treponema denticola* [29,30]. These bacteria are highly virulent and are involved in the destruction of periodontal tissue. The presence of these red-complex bacteria in periodontal pockets is strongly associated with severe forms. Another important bacterium, *Aggregatibacter actinomycetemcomitans,* has also been implicated in PD as a particularly aggressive form of the disease [31,32]. Additionally, recent studies have identified a potential role for other bacterial species in the progression and prognosis of periodontal. A second consortia of bacteria is the orange complex with different bacteria species present at the onset of PD [33]. One such bacterial species in the orange complex is *Fusobacterium nucleatum*, which has been shown to play a significant role in the formation of periodontal biofilm and its association with disease severity and AMR [32,34]. Recent studies have also highlighted the potential role of the oral microbiome, which encompasses all bacterial species present in the oral cavity, not just those directly associated with periodontal disease [35,36].

Overall, the role of bacteria in the prognosis of periodontal disease is complex and multifaceted. There is also growing evidence that the by-products produced by bacteria in the periodontal pocket may also play a significant role in disease progression and prognosis [37]. A study by Nakagawa and colleagues illustrated that long-term exposure to butyric acid, a bacterial metabolite causes the production of matrix metalloproteinase-8 (MMP-8) in excess and this when detected in oral samples can give an indication of progressive PD state [38]. Additionally, the presence of dental plaque and an imbalance in periodontal microflora promotes dysregulated oxidative stress leading to further injury and inflammation. There is proof that oxidative stress caused by reactive oxygen species (ROS) is periodontal tissue damage [39,40].

Future research in this area may focus on developing novel therapeutic approaches that target specific bacterial species or their by-products to improve the prognosis and treatment outcomes of PD.

## 3. Treatment Options Available for Treating Bacterial Infections in Periodontal Disease

PD can be prevented and possibly reversed by good and consistent oral hygiene procedures that remove microbial biofilm. These procedures include daily oral hygiene practices, gingival scaling, surgical procedures and antibiotic therapy. Mechanical oral hygiene practices such as flossing and toothbrushing are highly recommended and common effective methods for preventing PD and maintaining periodontal health [41,42].

Periodontal examination on a clinical visit is crucial to the choice of treatment in PD [2]. During such visits, an initial basic periodontal examination provides score values for a treatment plan. This plan may be for either non-surgical periodontal therapy, dental prophylaxis, supportive therapy or surgical interventions [6]. Non-surgical therapy involves encouraging adequate personal oral hygiene of patients and effective removal of calculus, stains and supra-gingival plaque deposits by using hand instruments such as a brush or a powered ultrasonic scaler. According to the SDCEP, a thorough review of available evidence indicates that debridement achieved by using powered scalers is not more effective than using hand instruments, rather, both methods are time consuming [6]. The effectiveness of non-surgical therapy is largely dependent on the patient achieving good personal oral hygiene.

To eliminate disease-associated pathogens in PD and ameliorate clinical condition, empiric antibiotic treatments are often used in combination with mechanical debridement [43]. Various antimicrobial agents are either administered orally or incorporated in chemical agents such as mouthwashes and toothpastes to inhibit plaque-causing bacteria. Antibacterial therapy is only appropriate for PD where there is spread of infection or evidence of systemic involvement [12].

Local antimicrobial treatments such as disinfectants and locally delivered antibiotics have been proposed as an effective way to treat PD [44]. These treatments can be used as a stand-alone therapy or as an adjunct to root surface instrumentation. Examples include anti-plaque agents in the form of mouthwashes, oral sprays, lozenges, chewing gums, dental floss, dentifrices, [45] toothpastes and other marketed products such as propolis [46,47]. A key antimicrobial agent in these formulations is chlorhexidine, usually as a gluconate [48].

By using local antimicrobials, odontogenic bacteria can be eliminated, leading to improved oral health. In the case of aggressive periodontitis and necrotising ulcerative gingivitis, in patients with comorbidities and in the case of poor response to conventional mechanical therapy, antibiotic therapy provides an effective treatment outcome.

The emergence and rapid spread of antibiotic resistance is a threat to public health. A worrying case of AMR was reported by Maestre and colleagues in 2007. In their aim to investigate the susceptibility of clinical periodontitis bacteria isolates to commonly prescribed antibiotics for PD prophylaxis and treatment, an alarming resistance rate from 18.2 to 47.7% to Azithromycin was recorded [49]. In addition, more than 50% *Prevotella species* isolated from the clinical samples were beta lactamase-producing organisms with 17.1 to 26.3% resistance to amoxicillin; fortunately, these species were susceptible to amoxicillin-clavulanic acid [49]. Among the three *Prevotella* species studied for their antimicrobial resistance to first- and second-line antibiotics in PD treatment, in an in vitro study, a key aspect of the findings revealed that *P. nigrescens* had the highest amoxicillin, amoxicillin clavulanic acid and clindamycin [50]. In a recent retrospective study by Jepsen et al., 2021, an exponential increase in the resistance to selected antibiotics with associated with a statistically significant decline in susceptibility to prescribed antibiotics was observed [51]. AMR is not only limited to oral antibiotics, but also to antiseptic agents such as chlorhexidine. What stands out in the work by Saleem et al., 2016, is the high rate of resistance of oral bacteria to multiple antibiotics and undiluted commercial chlorhexidine mouth wash [52]. Their study highlights the significance of heightened vigilance of multi-drug resistance bacteria and PD.

Despite the significance of antibiotics in PD treatment, the drug prescribing for dentistry guidelines, has highlighted one of the key drivers of antibiotic resistance in primary care and dentistry to be the indiscriminate use of antibiotics. In the NHS primary care data for Scotland, dental prescription of antibiotics accounts for 9% of oral antibiotics. In a recent clinical audit, about 50% of dental antibiotic prescriptions are inappropriate [12].

A rational antimicrobial treatment, routes of drug delivery and defined treatment goals are essential in enhancing the effectiveness of therapy in PD. Prolonged antibiotic therapy can encourage the development of drug-resistant microbes; hence, the need for surveillance on the selection and prescribing of antibiotics in PD therapy. Table 1 shows a list of commonly prescribed antibiotics in PD, their medicinal forms and AMR status.

## 4. Hydrogel Formulations

Previous studies mostly describe hydrogels as soft three-dimensional networks of crosslinked polymers that can accommodate large volumes of water into the polymeric network. This crosslinked network prevents the hydrogel unit from dissolution in spite of its high water affinity [63,64].

Hydrogels are classified according to a number of parameters, such as the source of polymers, technique of fabrication, crosslinking process as well as their response to environmental stimuli [65,66]. They are further categorised under these classes to give a better appreciation of their physicochemical characteristics and biomedical applications. Figure 2 illustrates the classification of hydrogels below.

Hydrogels as antibacterial agents can be amenable alternatives to conventional antibiotic therapies [67]. Depending on the type and fabrication, a broad range of hydrogels provide additional advantages to traditional antibiotic treatments [67,68]. These pros include local administration, prolonged and controlled release, targeted delivery and stimulated release. Hydrogels with antibacterial properties can be applied in the field of diagnostics, where for instance, formulations are used as coatings for catheters and surgical instruments [69]. Hydrogels can be inherently antimicrobial, preventing the adhesion of microorganisms due to their chemical composition [70,71,72,73]. Antibacterial hydrogels can be useful as wound-dressing materials, incorporated into bandages, in urinary tract infections, in contact lenses, as coatings for medical devices, in bone-related infections, in neonatal sepsis and as matrices in regenerative medicine [22,74].

Current research on antibacterial hydrogels includes biocompatible polymers such as polyethylene glycol of varied molecular weights, polysaccharides (such as chitosan, agarose and gelatin) and polyvinyl alcohol or compounds such as poly (methyl vinyl ether/maleic acid) with or without antibiotic substances [70,71,74,75,76,77,78]. The underlining basis for selecting hydrogels as antibacterial therapeutic agents are biodegradability and biocompatibility.

Bactericidal and bacteriostatic chemicals and compounds incorporated into hydrogels include nanoparticles (silver, copper) and antibiotics (Gentamicin, amoxicillin, vancomycin) [68,74,79,80,81]. Silver nanoparticle hydrogels are limited in their application due to their dose-dependent activity and unsatisfactory biocompatibility. Other antibacterial agents have limited use due to the evolution of drug-resistant bacteria. However, due to the tunable properties of hydrogels, the fabrication of smart antibacterial hydrogels and the surveillance of their antimicrobial properties, they will be vital in combating antimicrobial resistance.

Hydrogels that have inherent antibacterial properties could be applied as coatings for surgical or diagnostic instruments in periodontal disease [82]. This could potentially reduce the risk of infections and complications associated with these procedures, leading to better patient outcomes. Furthermore, hydrogels could also be used as a platform for developing combination therapies, where multiple antibiotics or other therapeutic agents can be incorporated into a single system for improved efficacy against infections. This concept is highlighted in an illustration (see Figure 3) where there is the incorporation of nanoparticles and antibiotics into the hydrogel matrix.

Hydrogels are termed smart based on their response to external stimuli, such as temperature, pH, light, ROS or electric fields, which triggers a reversible change in the hydrogel’s structure and properties [40,84]. For example, in thermo-responsive hydrogels, the layout of the polymeric network changes above a particular temperature known as the low critical solution temperature (LCST). At temperatures below the LCST, hydrophobic regions aggregate and the polymeric structure shrinks; hence, some hydrogels are formed above their LCST to obtain a gel state and below their LCST for a sol state see Figure 4A [85]. These hydrogels are liquid at room temperature and gels at human body temperatures. Synthetic polymers such as poly (N-isopropylacrylamine) (PNIAM), poly (N-vinylcaprolactam) and Pluronic are among the thermo-responsive polymers for the fabrications of hydrogels for drug delivery based on their LCST [85,86,87]. Pham and colleagues developed a thermo-responsive hydrogel using Pluronic F127, silk fibrin and methylcellulose for the delivery of metronidazole in periodontitis treatment using the LCST [88]. Other thermo-responsive hydrogels are formulated based on another critical temperature known as the upper critical solution temperature (UCST) (Figure 4B). For such hydrogels, their transition temperatures are below room temperature to maintain the integrity of the gel state. Polymers of acrylic acid derivatives and acrylamide derivatives are good candidates for these smart hydrogels.

Another category of smart hydrogels for antibiotic therapy in PD is pH-responsive hydrogels. pH-responsive hydrogels alter their swelling dependent on changes in pH. Key characteristics of pH-sensitive hydrogels are the presence of a cationic moiety such as the amine group or an ionic group such as carboxyl acid. The hydrogels are either anionic or cationic. Figure 4C,D illustrate the swelling of anionic and cationic hydrogels at different pH levels. Natural polymers such as gelatin, chitosan and alginate are fabricated as pH-responsive hydrogels [89]. Synthetic polymers for pH-responsive hydrogels include polyacrylamide, polyacrylic acid and poly diethylaminoethyl methacrylate. In PD, chitosan-based smart hydrogel systems have shown immense promise in enhancing PD treatment and periodontal drug delivery [90,91,92,93].

The swelling nature of chitosan hydrogels is dependent on factors such as the pH of the swelling medium, hydrophilicity, polymer concentration, ionic charges and ionizable groups, among others. In a cationic polymer such as chitosan, the protonation of the amino group favours swelling in an acidic medium. On the other hand, for carboxymethyl chitosan, an anionic polymer, a medium with higher pH favours its swelling due to the ionization of acidic groups [94,95].

Aycan and Alemdar developed a chitosan-based hydrogel reinforced with bone ash for the delivery of amoxicillin [96]. Specifically, poly-ethylene-glycol-diacrylate and chitosan-grafted glycidyl-methacrylate were photopolymerised under ultraviolet light to produce the hydrogel. The release of amoxicillin from the hydrogels at 37 °C was profound (12.64 mg drug per 1 g of polymer) in the acidic elution medium (pH 1.2) compared to results obtained at pH 7.4 (5.97 mg drug per 1 g polymer). The incorporation of bone ash did not significantly interrupt drug release. However, the release pattern for bone ash- containing chitosan hydrogels was controlled compared with the negative control.

**Figure 4 antibiotics-12-01073-f004:**
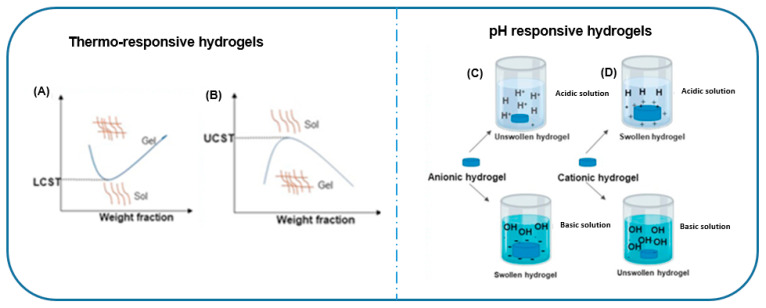
Schematic illustration of the sol-gel transition state for (**A**) lower critical solution temperature (LCST) and (**B**) upper critical solution temperature of thermo-responsive hydrogels. The swelling behaviour of (**C**) anionic and (**D**) cationic pH-responsive hydrogels. Adapted from [67].

Because smart hydrogels can release drugs in response to specific stimuli, allowing for targeted drug delivery and reduced side effects associated with overt exposure to active pharmaceutical ingredients in conventional drug-delivery methods [84,97]. This technique of using smart hydrogels has been used for the drug delivery of various antibiotics, including but not limited to penicillin, tetracycline, gentamicin and ciprofloxacin [98,99].

Kang and co-workers [99] designed a thermo-responsive antibacterial hydrogel using N-isopropylacrylamide, graphene, and viny carboxymethyl chitosan incorporated with ciprofloxacin. The antimicrobial activities of hydrogel formulations only and drug-loaded hydrogels were evaluated in vitro and compared. The results of zones of bacteria (*E. coli* and *S. aureus*) inhibition by ciprofloxacin-loaded thermo-responsive hydrogel formulations are exhibited in Figure 5A below. The ZOI for both bacteria ranged from 18.06 mm to 20.62 mm. When the bacterial mortality of unloaded and ciprofloxacin-loaded hydrogels was compared, drug-loaded hydrogels produce higher mortality rates 98.65% for *E. coli* and 94.71% for *S. aureus.* See Figure 5B [99].

An added advantage of using smart hydrogels for the delivery of antibiotics, in particular, is that they can provide sustained release of the antibiotic, ensuring a consistent and controlled dosage over an extended period of time. This can help to prevent the development of antibiotic-resistant strains of bacteria, as a constant and controlled dosage can ensure that all bacterial cells are effectively eliminated. Furthermore, using smart hydrogels for antibiotic delivery can also increase patient compliance. Depending on the fabrication, it removes the need for frequent dosing and reduces the risk of missed doses. In a study reported by Lin et al., a smart hydrogel that contained mesoporous silica on gold nanobipyramids was fabricated for the sustained release of light-thermal antibacterial and drugs against endodontic pathogenic bacteria [100] The release trend of minocycline, a tetracycline from the hydrogels was studied over a period of seven days. The results revealed a burst release over the first two days and a slow release over the subsequent 5 days. When two different release media are compared, the rate of minocycline released was enhanced in the collagenase solution compared to the phosphate-buffered saline. To reduce the risk of AMR associated with the traditional delivery of antibiotics, a photothermal antibacterial treatment approach was used against *Porphyromonas gingivalis* without compromising normal healthy tissues. The results indicate that hydrogels can deliver antibiotics over a prolonged period with approaches tuneable techniques to mitigate AMR. Table 2 highlights additional smart hydrogel formulations with antibacterial activity against periodontal pathogens.

An opposing argument could be that the sustained release of antibiotics through smart hydrogels may result in the development of antibiotic resistance, especially sustained suboptimal doses, are delivered [101]. The prolonged exposure to low doses of antibiotics can promote bacterial adaptation and mutation, leading to reduced effectiveness or complete ineffectiveness of the antibiotic over time. The sustained release may not address immediate infection symptoms as quickly as a higher dosage delivered at once would [102,103]. Therefore, optimizing the design and fabrication of smart hydrogels for antibiotic drug delivery is crucial to address these challenges. With this in mind, multi-layered smart hydrogels (MSHs) with varying degradation rates and drug release profiles have been proposed as a potential solution to these challenges [104]. MSHs have independent and controllable drug release layers with adjustable properties. These MSHs can provide an initial burst release of the antibiotic for immediate symptom relief, followed by sustained release to ensure consistent and controlled dosage over an extended period of time. MSHs were first fabricated in 2008 by Ladet et al. using chitosan and alginate [105]. Hyaluronic acid and carboxymethylcellulose are good candidates for MSHs [106,107]. Various techniques can be used to fabricate MSHs. These include three-dimensional bioprinting [108], a layer-by-layer approach [109], a self-assembly process [110] and sequential encapsulation [111].

Although smart hydrogels have great potential for antibiotic delivery, there are some concerns. The main challenges are achieving a high drug-loading capacity while maintaining the hydrogel’s stability and responsiveness and also ensuring that the antibiotic’s release kinetics are consistent and predictable. Additionally, the biocompatibility and toxicity of smart hydrogels must be thoroughly evaluated to ensure their safety for use.

**Table 2 antibiotics-12-01073-t002:** Smart hydrogels with antimicrobial activity against periodontal pathogens.

Type of Smart Hydrogel	Hydrogel Materials	Antibacterial Agent	Antibacterial Activity	References
Thermo-responsive hydrogel	N-isopropylacrylamide (NIPAM), silane-coupled with graphene (GM), vinyl carboxymethyl chitosan (CG); NIPAM-CG/GM loaded with ciprofloxacin	Ciprofloxacin	Antibacterial activity against *E. coli* (ZOI 18.10 mm–20.16 mm ± SD) and *S. aureus* (ZOI 18.06 mm–20.62 mm ± SD)	[99]
Thermo-responsive hydrogel	β-glycerolphosphate disodium salt pentahydrate (β-Gp), chitosan, and gelatin	Metronidazole	Antibacterial activity of hydrogel–metronidazole formulation with MIC of 40 µg/mL for *Clostridium sporogenes.*	[112]
Reactive oxygen species (ROS) responsive hydrogel	PVA and N1-(4-boronobenzyl)-N3-(4-boronophenyl)-N^1^, N^1^,N^3^,N^3^-tetramethylpropane-1,3-diaminium	Macrophages (derived from bone marrow) and C5aR antagonist	Blockage of C5aR, a target site controlled by periodontal bacteria Pg (*Porphyromonas gingivalis*)	[113]
Thermo-responsive hydrogel	Chitosan, glycerophosphate salt	Chitosan	Incorporation of atorvastatin and lovastatin reduced Porphyromonas *gingivalis*-induced inflammation	[114]
Hybrid hydrogel	β-cyclodextrin, ciprofloxacin, polydioxanone,	Ciprofloxacin	Increased antimicrobial activity of drug-loaded hydrogel against endodontic pathogen (*Enterococcus faecalis*) compared to 0.12% chlorhexidine	[115]
Thermo-responsive hydrogel	Pluronic F127, silk fibrin, and methyl cellulose,	Metronidazole	A 200-fold of reported MIC (0.063–0.514 µg/mL) of metronidazole for *Porphyromonas gingivalis* was released from all formulations	[88]
pH-sensitive hydrogel	Chitosan, sodium alginate, polyethylene glycol, 1,4 diaminobutane	Sodium ceftriaxone	Antimicrobial activity with ZOI of 10.2 mm and 9.7 mm for test organisms *S.aureus* and *E.coli* respectively	[93]
Photo-sensitive	Methacrylic gelatin, Methacrylic polyphosphoester	Zeolitic imidazolate framework	Bacteria (*Porphyromonas gingivalis* and *Streptococcus mutans*) biofilm destruction by hydrogels.	[116]
Near-infrared (NIR) light-sensitive hydrogel	Gold nanobipyramids, gelatin methacrylate, mesoporous silica	Minocycline	A 90% antibacterial efficacy against *Porphyromonas gingivalis*	[100]

### 4.1. Release Mechanism of Antibiotics by Hydrogels

Identifying the appropriate drug release model for the hydrogel is key. Responsive hydrogels may also provide a controlled drug release profile of the hydrogel [76]. Understanding the physiochemical characteristics of the drug and hydrogel is key in identifying the release of the drug. Additionally, even though this challenge is of concern, hydrogels can be tailor made by selecting specific polymers or monomers, crosslinkers and fabrication conditions to produce the final product of desired characteristics.

Drug release mechanisms can be broadly categorised as either rapid or sustained release. The drug release mechanisms of hydrogels rely heavily on the physicochemical properties of hydrogels. Drugs can be bound to hydrogels through a non-covalent interaction which allows drug molecules to move freely within the hydrogel network [117]. There are many parameters that can be adapted to modify the release rate from hydrogels but a critical one is the mesh size [118]. This parameter is controlled by the polymer and the crosslinker concentration. The relationship between the drug size and the mesh size will influence the release rate. When the drug is smaller than the mesh size, the cargo molecules can move and diffuse freely within the hydrogel [118]. On the other hand, when the drug and the mesh size present similar sizes the release rates depend not only on the diffusivity of the drug molecule but also on the steric hindrance on drug diffusion. This can contribute to slower release rates. If the drug cargo is larger than the mesh size (this can be observed for macromolecules) the drug cargo will be trapped inside the hydrogel structure [118]. Another alternative to enhance the hydrogel–drug interaction is by modifying the chemical nature of the hydrogel to allow non-covalent interactions between the drug and the hydrogel structure. This can be achieved by including molecules such as cyclodextrin to form inclusion complexes or just by adding hydrophobic moieties that can lead to interactions with molecules to extend drug release [119,120,121,122]. Additionally, there are other parameters affecting drug release from hydrogels such as the drug loading. When the hydrogel is administered, a concentration gradient between the drug and its immediate surrounding is formed where there is an initial upsurge release of drugs from the hydrogel [117,123]. Over time, this rapid release of the drug from the hydrogel slows down. In instances where rapid release is essential to positive treatment outcomes, this mechanism is vital. On the other hand, the rapid release of some drugs is in tandem with untoward side effects. Because of the latter, drug molecules are physically or covalently attached to hydrogels at the gelation stage. This way, through stimuli responses, the hydrogel can effectively regulate the release of drugs to target sites.

The antimicrobial action of smart hydrogels could be triggered using biological (enzymatic action or pH changes due to the presence of bacteria) or non-biological (light, temperature, magnetic and electric field) stimuli [82].

### 4.2. Feasibility of Incorporating Antibiotics into Hydrogels

Due to the desirable characteristics of hydrogels for drug delivery, hydrogels are potentially feasible as matrixes for antibiotics. Dias and team have successfully developed a novel approach to PD treatment by incorporating oxytetracycline in an advanced polymeric system [124]. Similarly, Forero-Doria et al. successfully fabricated a two-polymer content hydrogel with microcrystalline cellulose and chalcone for the simultaneous sustained release of antimicrobial agents. In their formulation, allantoin, dexpanthenol, linezolid, and resveratrol were analysed for their drug release, cytotoxicity wound healing and antibacterial properties [76].

Hydrogels are good drug-delivery platforms for antibacterial agents that target the inhibition of bacteria biofilm [125,126]. Antifouling agents, antibiotics and quorum-sensing agents can be successfully incorporated into the hydrogel matrix. Bacteriophage encapsulating hydrogels have also been explored as a novel strategy for the delivery of antibacterial agents to target bacteria biofilm. Injectable hydrogels fabricated using adhesive peptides as starting materials have been designed to treat local bone infections by targeting bacteria biofilm. The injectable hydrogels are capable of encapsulating bacteriophage and the delivery the active phage at the site of bone infection. In comparison with bacteriophage-free injectable hydrogels, the bacteriophage-encapsulating hydrogels reduced live bacteria count by a quadruple factor 7 days post implantation [126].

Overall, advantages of hydrogel-infused antibiotics include: potential advantage of localised treatment [76], advantage of hydrogels co-infused with more the one bio-compatible antibiotic or antibacterial agent, potential shorter duration of action and the potential to avoid the side effects of oral antibiotics.

## 5. Potential Roadblocks

Although biocompatible hydrogels are appreciated for their high-water content, their ability to hold approximately 70% water in their polymer network is a limitation to their mechanical properties and by extension their applications. There is, therefore, the need to initially establish a database of the target product profile of the intended biomedical device for which the hydrogel is to be used for. Following this, a library of initial starting materials for the hydrogels, fabrication techniques and parameters and physicochemical characterisation are to be considered in developing suitable hydrogel materials for the intended use. For PD treatments where hydrogels are used as drug delivery of antibiotics, considerations such as potential greater offset to the oral microbiome compared to conventional oral therapies, taste effects (palatability), and fine-tuning exposure kinetics of the hydrogel device loaded with antibiotics are crucial in positive overall patient experience and treatment outcomes. Drug release models such as Higuchi, Weibull function and Korsmeyer–Peppas could be adopted to enhance the understanding of the release kinetics of hydrogels.

## 6. Limitations of this Review

First, as this paper is a narrative review, a number of articles may not have been identified and included. Secondly, because this paper attempts to provide details of key considerations for hydrogels in PD diagnostics and management, in-depth discussions of some sections are limited to the search strategy used. However, this paper highlights a heterogenous source of information on PD, strategies for antibacterial treatment using hydrogels and key factors for consideration in POCT, thus assuring the appropriate level of comprehensiveness.

## 7. Future Directions

This review underscores the need for a multidisciplinary approach to the treatment of periodontal infections, combining innovative technologies and materials with traditional periodontal therapy techniques. Collaboration between dental professionals, researchers and material scientists can facilitate the development of more effective and personalized treatment plans for patients suffering from periodontal infections, ultimately improving the overall standard of care in the field. Furthermore, public health initiatives aimed at promoting oral hygiene and preventative care can play a critical role in reducing the incidence of periodontal infections and the need for antibiotic treatment altogether, further contributing to the fight against AMR. A comprehensive and integrated approach is necessary to effectively address the issue of periodontal infections and antibiotic resistance.

## 8. Conclusions

Hydrogel-based antibiotic drug-delivery devices have shown promising potential for the treatment of periodontal infections, offering targeted and sustained release of antibiotics. Further research and development in this field could lead to the creation of more effective treatment options for periodontal infections that can ultimately improve patient outcomes and quality of life, while addressing the growing issue of antibiotic resistance. Therefore, it is important to continue exploring the potential of hydrogel-based antibiotic drug-delivery devices and other alternative treatment options for periodontal infections to combat the rise of antibiotic resistance and improve overall oral and systemic health.

## Figures and Tables

**Figure 1 antibiotics-12-01073-f001:**
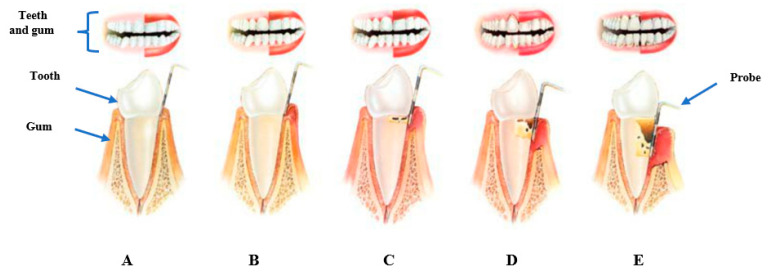
Schematic diagram of the progression of PD (**A**) represents healthy teeth and gum characterised by pink gums with fresh breath and no bleeding with brushing. (**B**) Gingivitis marked by red swollen gums with bleeding during brushing, possible bad breath, no bone loss. (**C**) Early stages of periodontitis, with red swollen gums that bleeds during brushing, mild bone loss, possible tooth mobility and bad breath. (**D**) Moderate periodontitis with red swollen bleeding gums, persistent bad breath, moderate bone loss, tooth mobility and root exposure. (**E**) Severe periodontitis characterised by red swollen gums that bleeds with brushing, advanced bone loss, possible tooth loss and persistent bad breath. Image adapted from [7].

**Figure 2 antibiotics-12-01073-f002:**
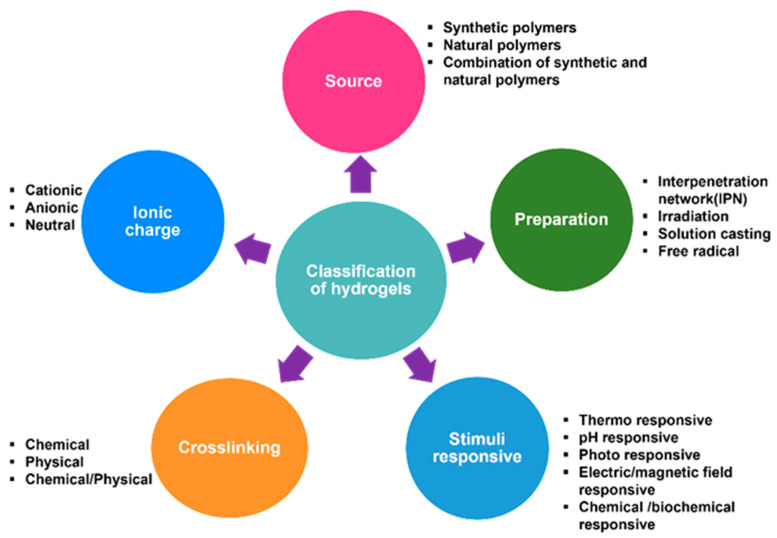
Classifications of hydrogels.

**Figure 3 antibiotics-12-01073-f003:**
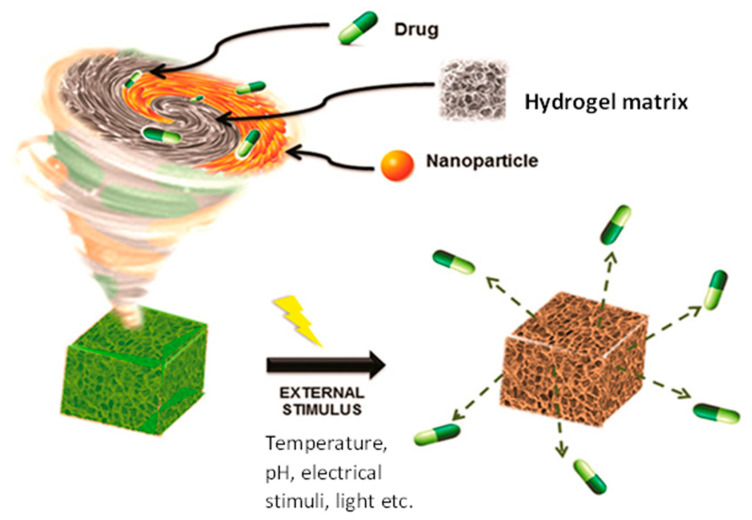
Illustration of an incorporation of antibiotic and antibacterial nanoparticles into hydrogel matrix for drug delivery in response to external stimuli. Image adapted with permission [83].

**Figure 5 antibiotics-12-01073-f005:**
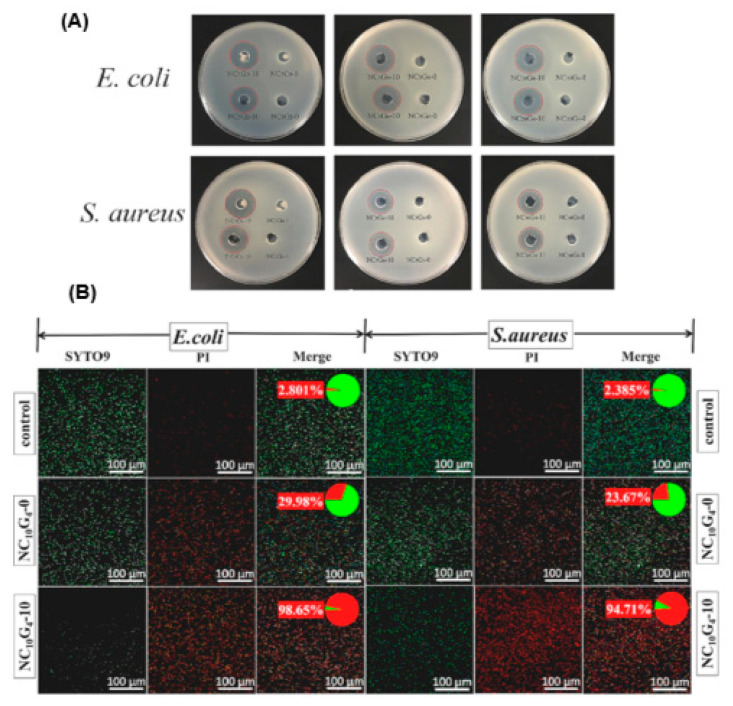
(**A**) ZOI (top—*E. coli*, bottom—*S. aureus*) by ciprofloxacin-loaded hydrogel formulations. (**B**) Bacteria mortality post 24 h hydrogel treatment. Numerical values expressed as percentages represent bacteria mortality. Red stain indicates dead bacteria, and green stain represents live bacteria. Reproduced from [99]. Copyright 2023, Elsevier.

**Table 1 antibiotics-12-01073-t001:** Examples of common antibiotics prescribed in dental practice.

Generic Drug Name	Drug Class	Available Medicinal Form	PD Antimicrobial Resistance Reported	Formulation	Reference
Amoxicillin	Penicillins	Capsule, oral suspension, oral powder	Yes	Immediate release	[49,53,54,55]
Azithromycin	Macrolides	Capsule, eyedrop, tablet, oral suspension, powder for solution for infusion,	Yes	Immediate release	[49,56,57]
Cefalexin	Cephalosporins	Tablet, capsule, oral suspension	Yes	Immediate release	[58]
Chlorhexidine	Antiseptic and disinfectant	Oral gel, mouthwash, spray	Yes	Immediate release	[52]
Clarithromycin	Macrolides	Tablet, granule, oral suspension, powder for infusion	Yes	Immediate and modified release	[53,56,59]
Clindamycin	Lincosamide	Capsule, gel, cream, infusion, solution for injection, oral suspension	Yes	Immediate release	[54,55,57,60,61]
Co-amoxiclav	Penicillins	Tablet, oral suspension, powder for solution for injection	No	Immediate release	[49,61]
Doxycycline	tetracyclines	Capsule, tablet, dispersible tablets	Yes	Immediate and modified release	[51,55]
Erythromycin	Macrolides	Tablet, oral suspension, powder for infusion	Yes	Immediate release	[62]
Metronidazole	Nitroimidazole derivative	Tablet, oral suspension, infusion, suppository, cream, gel	Yes	Immediate release	[51,58]
Oxytetracycline	tetracyclines	Tablet, oral suspension	-	Immediate release	[53]

## Data Availability

Not applicable.

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
