# Peer review of "Treatment of Periodontal Infections, the Possible Role of Hydrogels as Antibiotic Drug-Delivery Systems"

_antibiotics, 2023, doi:10.3390/antibiotics12061073_

Round 1
Reviewer 1 Report
The current manuscript is an interesting review on the use of hydrogels for the treatment of periodontal infections. Although the authors do a good job in describing several introductory aspects, the most important part, which should be about the studies that have been done with the development of the hydrogels themselves, is lacking, making this a short and incomplete article. Authors only talk about hydrogels in 4 out of 15 pages, and in a superficial and “definition-like” manner. Authors should provide more specific information on the included studies, such as type of hydrogel and specific composition, drug, type of bacteria that they were used on, and specific in vitro and/or in vivo results, with representative images (taken from the original articles, given the right permission). Only then will it be truly interesting for the readers of a Q1 journal such as Antibiotics.
Reviewer 2 Report
Dear authors
Below you will find some suggestions that seek to enrich your work so that it is published as you expect, but that it has the scientific quality that this journal deserves.
Within numeral 2, it is important that this part be expanded a little since recent publications have implicated other microorganisms in the progression of this disease and related to the new classification of periodontitis, additionally taking into account pathobionts such as Prevotella that despite being normal microbiota is also associated with periodontitis and its resistance is clearly known.
I suggest including a more up-to-date reference of the part referring to what is proposed in lines 183 to 185 (Castillo, Y., Delgadillo, N. A., Neuta, Y., Hernández, A., Acevedo, T., Cárdenas, E., Montaño, A., Lafaurie, G. I., & Castillo, D. M. (2022). Antibiotic Susceptibility and Resistance Genes in Oral Clinical Isolates of Prevotella intermedia, Prevotella nigrescens, and Prevotella melaninogenica. Antibiotics (Basel, Switzerland), 11(7), 888. https://doi.org/10.3390/antibiotics11070888)
Despite being a literary review with a lot of information, it does not leave the main idea, which is why can hydrogens with antibiotics be a good alternative for the treatment of periodontitis? Data on antimicrobial efficacy should be included in the literature. There are recent data on this effect in vitro against bacteria of importance in periodontal disease.
Although they include an item of limitations, I consider that these could be less if more recent references are included, and with the data of the results of the antimicrobial effect they can improve perhaps, they can improve the search strategy to update references and include relevant data.
Round 2
Reviewer 1 Report
Manuscript has been quite improved.
Reviewer 2 Report
I believe that with the changes made the paper is more solid.
The only observation to correct is
Line 191: typo P. nigricans is P. nigrescens
